# Ophthalmological Manifestations in Inflammatory Bowel Diseases: Keep an Eye on It

**DOI:** 10.3390/cells13020142

**Published:** 2024-01-12

**Authors:** Giulia Migliorisi, Giovanna Vella, Arianna Dal Buono, Roberto Gabbiadini, Anita Busacca, Laura Loy, Cristina Bezzio, Paolo Vinciguerra, Alessandro Armuzzi

**Affiliations:** 1IBD Center, IRCCS Humanitas Research Hospital, Via Manzoni 56, Rozzano, 20089 Milan, Italy; giulia.migliorisi@humanitas.it (G.M.); arianna.dalbuono@humanitas.it (A.D.B.); roberto.gabbiadini@humanitas.it (R.G.); anita.busacca@humanitas.it (A.B.); laura.loy@humanitas.it (L.L.); cristina.bezzio@hunimed.eu (C.B.); 2Department of Biomedical Sciences, Humanitas University, Via Rita Levi Montalcini 4, Pieve Emanuele, 20072 Milan, Italy; paolo.vinciguerra@hunimed.eu; 3Department of Ophtalmology, IRCCS Humanitas Research Hospital, Via Manzoni 56, Rozzano, 20089 Milan, Italy; giovanna.vella@mc.humanitas.it

**Keywords:** uveitis, episcleritis, scleritis, inflammatory bowel disease, treatment

## Abstract

Background and aims: Inflammatory bowel diseases (IBD) are multifactorial chronic inflammatory disorders affecting the gastrointestinal tract. However, a broad spectrum of extraintestinal manifestations (EIMs) is associated with IBD, affecting several organs and systems, such as the skin, musculoskeletal and hepatobiliary systems, and, not least, the eye. Approximately 10% of IBD patients can develop ocular EIMs (O-EIMs) with a higher prevalence in Crohn’s disease (CD). Eye-redness, photophobia, pain, and blurred vision are the common symptoms, with a wide rate of severity and clinical impact on the quality of life. This narrative review aims to summarize the prevalence, pathogenesis, and current evidence-based management of O-EIMs, underlying the importance of a holistic approach and specialties collaboration for a prompt diagnosis and treatment. Methods: PubMed was searched up to December 2023 to identify relevant studies investigating the pathogenesis, epidemiology, and treatment of O-EIMs in IBD patients. Results: The mechanisms underlying O-EIMs are partially unknown, encompassing immune dysregulation, shared antigens between the eye and the gut, genetic predisposition, and systemic inflammation driven by high levels of interleukins and cytokines in IBD patients. The complexity of O-EIMs’ pathogenesis reflects in the management of these conditions, varying from topical and systemic steroids to immunomodulatory molecules and biologic therapy, such as anti-tumor necrosis factor (TNF)-alpha. A multidisciplinary approach is the backbone of the management of O-EIMs.

## 1. Introduction

Inflammatory bowel diseases (IBD), including ulcerative colitis (UC) and Crohn’s disease (CD), are complex multifactorial chronic inflammatory disorders affecting nearly 7 million patients worldwide, with an increasing prevalence in Northern Europe and emerging countries [1]. IBD can be associated with a wide spectrum of extra-intestinal manifestations (EIMs) with a significant impact on patients’ quality of life [2]. The current definition of EIMs encompasses inflammatory processes associated with IBD’s activity and flares, whether dependent or independent [3,4,5]. The prevalence of EIMs varies from 6% to 47% [5]: a recent systematic review and meta-analysis that examined 52 studies revealed that almost one quarter of IBD patients experienced at least one musculoskeletal, ocular, or skin EIM [4]. Additionally, these manifestations can present prior to IBD diagnosis in up to 25% of the patients [6]. EIMs seem to be more common in CD than UC patients [6,7], and the presence of a singular EIM predisposes to develop a new immune-mediated manifestation in another district [8]. The most commonly affected systems are the joints (i.e., arthritis and enthesitis), the skin (i.e., pyoderma gangrenosum, erythema nodosum, aphthous stomatitis), and the ocular and hepatobiliary districts (such as primary sclerosing cholangitis). EIMs’ underlying mechanisms and pathogenesis are poorly understood; they can either be reliant on or independent of intrinsic intestinal inflammation activity and share common activated immune molecular pathways [9]. An extension of gut inflammation may provide a plausible explanation; for example, gut chemokines, involved in T cell trafficking and migration into the bowel, can be expressed in biliary ducts and are associated with primary sclerosing cholangitis (PSC) in IBD patients [10]. Additionally, an uncontrolled auto-reactive T cell population, namely T-helper (Th1) by producing interferon-gamma (INF-Y) and interleukin 12 (IL-12) and T-helper 17 (Th17) with the release of interleukin 12 (IL-12) and interleukin 17 (IL-17), plays a crucial role in the pathogenesis of EIMs. Thus, patients affected by both IBD and non-infectious uveitis presented high levels of these cytokines [11]. Furthermore, IBD patients may have an intrinsic propensity towards developing independent autoimmune disorders, such as a genetic predisposition, which potentially explains the occurrence of EIMs during IBD remission. Indeed, certain haplotypes, such as HLA-DRB1*0103 or HLA B*58, are reported to be involved in musculoskeletal and eye manifestations [5]. Finally, dysbiosis and the selection of specific gut commensals, with the loss of diversity, have also been linked with the onset of various EIMs [12].

The prevalence of ocular extra-intestinal manifestations (OEIMs) in patients with IBD ranges from 0.3% to 13% [13]. These O-EIMs are mostly represented by episcleritis, scleritis, anterior uveitis, and iridocyclitis. Eye redness and pain or blurred vision are the most common symptoms. Episcleritis is the most frequent manifestation (2–5%), being mostly benign and strongly related to intestinal inflammatory flares, while uveitis (0.5–3.5%) is a more severe condition, potentially leading to vision loss, and independent of IBD activity [14]. O-EIMs are more common in children [15] and women and are associated with arthritis and pyoderma gangrenosum in CD and UC patients, respectively [16]. Other conditions, such as vascular involvement, optic neuritis, papillitis, or myositis are rarer.

This narrative review aims to summarize and clarify O-EIMs’ characteristics and their clinical manifestation, with a particular focus on the molecular pathogenesis, as well to discuss current and future management perspectives.

## 2. Materials and Methods

The keywords ‘inflammatory bowel diseases’, ‘Crohn’s disease’, ‘ulcerative colitis’, ‘uveitis’, ‘scleritis’, ‘episcleritis’, ‘ocular extra-intestinal’, and ‘ophthalmological involvement’ were used to search for significant and relevant publications on the pathogenesis of ocular manifestations in patients with IBD and any related therapeutic applications on Pubmed until December 2023. This search collected animal models and clinical studies.

## 3. Pathogenesis

The pathophysiology of EIMs in IBD is intricate and uncertain. The European Crohn’s and Colitis Organization (ECCO) proposed an operational definition of EIMs’ pathology in IBD patients. These manifestations can either result from the translocation of intestinal inflammation or be completely independent of IBD activity, arising from a genetic predisposition to autoimmunity disorders or environmental factors [15]. IBD and EIMs are thought to be the result of the interplay of environmental factors, genetic predisposition, changes to the intestinal microbiota, and immune system dysregulation, which all ultimately cause damage to the mucosa [16,17].

The ultimate disarray of intestinal cells’ tight junctions and the change in the composition of the epithelial mucus film allow luminal antigens to enter bowel epithelial cells. Additionally, molecular pattern-recognition receptors (e.g., Toll-like receptors TLRs) interacting with commensal microbiota can induce the activation of dendritic cells and macrophages. Consequently, the activation of multiple signaling pathways and upregulation of proinflammatory gene transcription resulted in the increased production of proinflammatory cytokines and recruitment of leukocytes, which perpetuates inflammation [18,19]. Another potential pathogenetic mechanism is represented by the formation of circulating antigen–antibody complexes or the production of autoantibodies, directed against common shared antigens by the colon and other tissues, including the eye [20]. In particular, it has been suggested that system-wide inflammation could be caused by the local impact of antigen–antibody complexes generated against bowel walls’ blood vessels and conveyed through the bloodstream [14]. The resulting damaged intestinal barrier allows antigens and microorganisms to penetrate, leading to a reactive lymphoid tissue response. In their initial Mendelian randomization study [21], Meng et al. illustrated that specific cytokines produced in the bowel (namely IL-6, IL-10, and IL-17) may participate in the origin of iridociclytis (IC) and EIMs by circulating to the eye and triggering ocular inflammation (Figure 1), implying that there may be a shared immunopathogenesis among the gut and eye [22]. Moreover, as previously said, shared antigens in the gut and eye may contribute to inflammation in both locations for IBD patients. For example, peptide 7E12H12 is present in both colon epithelial and non-pigmented ciliary epithelial cells and could represent an inflammation target [23]. Additionally, several studies suggest that a human epithelial colonic autoantigen, shared by skin, bile ducts, eyes, and joint cells, may trigger an antibody-mediated immune response in ulcerative colitis (UC) patients [21,22,23,24]. A higher intestinal damage and permeability caused by transmural inflammation in CD, may be a potential explanation for the greater prevalence of o-EIMs in CD patients compared to UC ones. However, there is no strong evidence in the literature confirming this hypothesis, and further studies are warranted.

The pathogenesis of EIMs questions genetic predisposition. Individuals with IBD who have specific haplotypes, mainly *HLA-B27* and *HLA-DRB1*0103*, were reported to be at higher risk of developing an extensive gut disease and the emergence of EIMs, particularly ocular and articular ones [24,25]. In a significant retrospective study, Lin et al. [26] proposed that a family history of IBD could potentially lead to an independent and amplified likelihood of developing ocular inflammation, even when there are no known instances of bowel disease or genetic susceptibility (*HLA*-*B27*). However, Lanna et al. [27] analyzed 96 Brazilian IBD patients and did not establish any association between particular HLA and ocular or joint manifestations. This finding suggests that genetic heterogeneity across diverse global populations could explain the different outcomes. The development of IBD is assumed to be attributable to around 150 unique genetic mutations [13,28,29]. The majority of these genes are involved in the primary immune response, specifically related to the autophagic function of leukocytes and the production of pro-inflammatory cytokines and chemokines (e.g., *NOD2*/*CARD15*, *ATG16 L1* or IL23R). Additionally, genes involved in transmembrane signaling systems of intestinal epithelial cells can be involved (*HNF4A*, *GNA12*) [30,31]. Certain rare monogenetic alterations may contribute to the early onset of IBD [30]. Specific alterations of the Nod2 gene on chromosome 16, can induce an abnormal activity of the innate immunity in response to gut microbiota, leading to a higher risk of developing CD [32]. Furthermore, the mutation of *NODd2*/*CARD15* is associated with a systemic autoinflammatory granulomatous disease, Blau syndrome, characterized by uveitis, arthritis, and dermatitis [33]. However, Nod2 is just a minor component of the complex genetics involved in IBD and EIMs [34].

Environmental factors may represent additional pathogenic factors. Vitamin D deficiency could have a shared role in the development of IBD and ocular inflammation. Several studies have shown that vitamin D has an anti-inflammatory effect, achieved through the suppression of B and T cells’ proliferation and differentiation [35]. An inverse correlation has been established between vitamin D levels and the risk of developing IBD and iridocyclitis (IC), indicating that higher vitamin D levels play a protective role [21,36]. Low vitamin D levels, resulting from inadequate sunlight exposure, are risk factors for all types of uveitis (including IC), UC, and CD [37]. These results suggest that vitamin D supplementation should be considered as an option for the prevention of uveitis relapse in high-risk patients.

Impaired autophagy in macrophages may considered another potential pathogenetic mechanism. Therefore, autophagy seems to play a vital role in maintaining ocular immune homeostasis, as the deletion of autophagy genes has been associated with worsening ocular inflammation severity due to inflammasome-mediated IL1B secretion. In their study using a mouse model, Santeford et al. found that a particular polymorphism (*Thr300Ala* or *T300A*) of the autophagy gene *ATG16L1* was linked to an increased risk of developing both CD and uveitis, suggesting a potential connection between IBD and o-EIMs [38].

Additionally, gut microbiota and dysbiosis may contribute to the pathogenesis of o-EIMs through molecular mimicry, although current data are inconclusive. Therefore, gut commensals may have a dual role. On one hand, they can participate in autoimmune ocular processes. For instance, a 2015 study using mice models found that gut bacteria antigens directly activate auto-reactive T cells that are involved in autoimmune uveitis [9,39]. Conversely, particular species such as *Lactobacillus reuterii* can provide a protective function by strengthening gut intraepithelial lymphocytes (IELs) that regulate autoreactive T cells [40]. Moreover, the elimination of gut commensals was correlated with an attenuation of the severity of ocular inflammation [39].
Figure 1In Figure 1, diverse immune mechanisms that underlie O-EIMs are elucidated: ectopic ocular expression of gut-specific chemokines and adhesion molecules (gut-specific chemokines and adhesion molecules (i.e., MAdCAM-1, CCL25, CCR9); T cell trafficking driven by non-specific adhesion molecules (i.e., VAP-1, CXCR3, and CCR5); common peptide sequence shared by enteric bacteria and host molecules (in a mice-model study, retina-specific T cells involved in uveitis pathogenesis are activated in the gut by a microbiota-dependent activation [39]); circulating antibodies-antigens complexes or autoantibodies directed against shared epitopes could expand inflammation outside the gut (i.e., eye); genetic predisposition (specific HLA complexes have been associated with specific EIM); activated neutrophils and macrophages leads to uncontrolled innate immune response in non-intestinal districts.
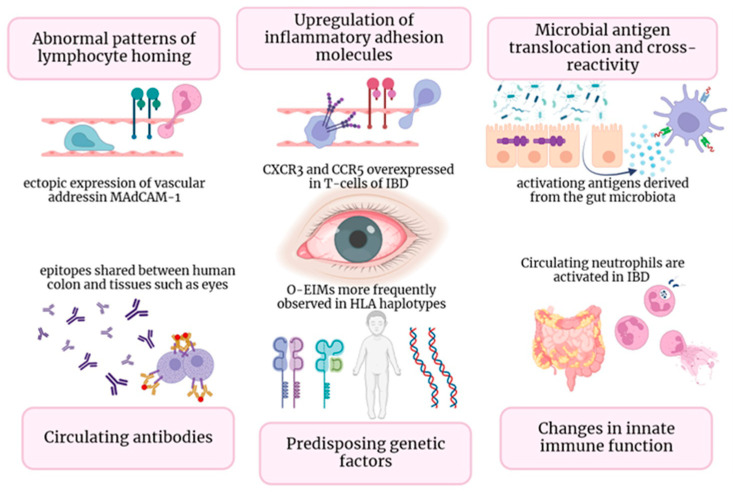


## 4. Episcleritis and Scleritis

### 4.1. Episcleritis

Episcleritis is a benign, non-granulomatous inflammation of the thin, richly vascularized layer between the sclera and the conjunctiva. In almost a third of cases, it is associated with a systemic autoimmune disorder [41,42] and it represents the most frequent O-EIMs in both CD and UC. A recent systematic review [17] found no difference in episcleritis frequency between UC and CD. The inflammation is mediated by the activation of resident immunity cells releasing cytokines and inducing vasodilation with vessels’ engorgement [42]. De la Maza et al. reported a potential role of interleukin-22 (IL-22) in the pathogenesis of episcleritis and scleritis [43,44]. IL-22 typically protects against bacterial and fungal infections, but recent findings suggest that it amplifies the effects of other cytokines that are known to be involved in ocular inflammation, such as interleukin 1-β (IL-1β), tumor necrosis factor-α (TNF-α), interleukin 6 (IL-6), and IL-17. Additionally, elevated levels of IL-22 were detected in patients with scleritis and episcleritis [42,44].

Episcleritis is self-limited and can serve as a marker for IBD activity, as it is more commonly linked with intestinal flares rather than quiescent phases [15,45,46]. It can affect one or both eyes, causing hyperemia, burning, itching, and mild discomfort and pain. No vision loss or change in pupillary response to light is reported [47]. Episcleritis can present as a diffuse inflammation impacting the entire layer or a sectoral formation, with a reddish nodule emerging from the episclera seen in 15% of patients [48]. No differences in terms of prognosis between the two forms are reported [13,42]. Episcleritis can be diagnosed by the instillation of a topical vasoconstrictor (10% phenylephrine eye drops) that brings enlarged blood vessels to blanch [49]. However, misdiagnosis and confusion with other ophthalmic conditions that also cause eye redness, like conjunctivitis and scleritis, are common. While the first one is a benign common manifestation with infectious (viral or bacterial) or traumatic causes and not related to IBD, the second one is a more severe condition that needs rapid specialist referral.

### 4.2. Scleritis

Scleritis is a sudden inflammatory condition of the sclera and its underlying vessels; at ocular examination they typically do not whiten after administration of vasoconstrictor injection, differently from episcleritis. It may lead to loss of vision and is usually rarer than episcleritis, affecting only 1% of IBD patients [14]; older age, female sex, and Caucasian ethnicity are potential risk factors [46,49,50]. Furthermore, individuals with Crohn’s disease are noted to have a 1.5-fold-heightened probability of experiencing scleritis compared to those with UC, as well as a greater possibility of recurrence [50,51]. In contrast to episcleritis, scleritis is more prevalent during IBD’s remission stage [15,46,47,49]. It is classified in anterior and posterior according to the anatomical insertion of rectal muscles, and, subsequently, in diffuse, nodular, and necrotizing forms [52,53]. Symptoms include eye redness and tenderness with blurred vision. Intense pain is present, especially at night, radiating to the head and jaundice, following trigeminal innervation. Anterior scleritis is macroscopically characterized by a violaceous scleral injection [54,55] and can be associated with anterior uveitis and ocular hypertension. Conversely, necrotizing scleritis with layer-thinning resulting in perforation is a rare and anecdotic event in IBD [55]. Nevertheless, it is a strong predictor of irreversible vision loss when occurring together with posterior uveitis and ocular [42]. Scleritis pathogenesis is not fully well known. To date, no specific genetic predisposition to scleritis has been identified, other than evidence that IBD patients in the same family are more likely to develop similar EIMs [56] and that specific haplotypes such as HLA-B7 have been associated with uveitis [57]. Autoantibodies (e.g., pANCA, cANCA) and an imbalance of matrix metalloproteinases (MMPs) can be involved [58], as well as a potential role of gut microbiota and dysbiosis [59].

## 5. Uveitis

Uveitis is defined as inflammation of the uveal tract, including the ciliary body, choroid, and iris, and is responsible for 5–10% of vision loss worldwide [60]. A recent systematic review and meta-analyses reported an uveitis prevalence of 2.38% in IBD patients. Uveitis was statistically significantly more prevalent in CD (3.27%) than in UC patients (1.60%, *p* < 0.05) [17,61]. While CU’s activity and uveitis do not correlate, uveitis is much more frequent and persistent during CD flares [62]. In addition, CD patients are more likely to develop bilateral forms [62] and an insidious onset [63], both of which are strong predictors of a higher clinical disease activity index with the need for immunomodulators or calcineurin inhibitors [64]. Anterior uveitis (iritis) is a non-granulomatous inflammation, the most common in IBD patients, presenting with acute pain, conjunctiva injections, photophobia, blurred vision and alteration of pupil [62,65]. Ophthalmological examination reveals keratic precipitates in the anterior chamber cell with posterior synechiae [49] (Figure 2). Intermediate (vitritis) and posterior uveitis are rarer [65,66]; they are high-threatening conditions that may lead to irreversible vision loss, retinal vein occlusion or retinal detachment [65]. In addition, isolated case reports show that posterior segment inflammation can lead to severe chorioretinitis [67]. O-EIMs clinical manifestation are represented in Figure 2.

The pathogenesis of uveitis is complex and not fully understood. As previously said, genetic factors may play an important role. For example, Orchard et al. reported a strong association of some human leukocyte antigens (*[HLA]-B27*, *B58*, and *HLA-DRB1*0103*) with uveitis and IBD [36]. Different mutations of nucleotide oligomerization domain containing the protein 2 gene (*NOD2/CARD15*) are associated with CD and may take part in the pathogenesis of uveitis in these patients [68].

However, non-infectious uveitis is currently considered to be a multigenic disease triggered by environmental factors, in which the gut microbiota may be of paramount importance [69]. The first confirmation of this hypothesis came in 2014 from Lin et al., who observed that transgenic mice, human HLA-B27 carriers, were at higher risk of developing uveitis and had a greater relative concentration of pro-inflammatory bacteria (*Prevotella* and *Streptococcus* species) in their gut microbiota than controls [70]. IBD gut dysbiosis may be involved in the pathogenesis of uveitis via several mechanisms; for instance, ocular inflammation is correlated with disruption of gut homeostasis and increased intestinal permeability in animal models [71]. Furthermore, a decrease in *Clostridium* and *Bacteroides* in IBD patients could be responsible for a reduction in de-hydroxylation of primary bile acids into secondary ones, which have an anti-inflammatory role and can prevent uveitis onset [72]. Furthermore, intestinal dysbiosis in IBD could reduce short-chain fatty acids (SCFAs), protective molecules produced by the gut microbiota from fiber fermentation, which normally enhance mucus production through immunomodulatory properties [73]. For example, a recent mice-model study found that the administration of SCFAs could attenuate uveitis in rats through stimulation of T cell regulators, concomitant suppression of T cell effectors and reduction in leukocyte migration from the gut to the eye [74]. In addition, the loss of the gut barrier and the resulting translocation of bacteria, toxins, and antigens are documented in many other EIMs, even if more data are needed [75].

## 6. Corneal Disease and Other Rare Manifestations

Corneal disorders in IBD were first described in the early 1920s by Crohn. He reported two IBD patients presenting with xerophthalmia and keratomalacia, which he attributed to vitamin A deficiency [76]. This suggests that IBD patients may experience corneal thinning and reduced tear production due to malabsorption of vitamins. In a prospective, case–control, and cross-sectional study, Barta et al. [77] suggested that CD patients rather than UC tend to develop dry eye disease (DED) as the ocular tear film characteristics are more impaired in CD patients than in UC patients. For this reason, DED is a relative contraindication to laser refractive surgery. Thus, two cases of necrotizing keratitis in CD patients have been reported after LASIK and PRK procedures. Both patients develop bilateral stromal inflammation, requiring prompt systemic and topical steroid treatment. Moreover, one of them experienced a relapse of the symptoms 7 months later [78]. Keratopathy rarely precedes IBD diagnosis: symptoms include ocular pain, foreign body sensation, sensitivity to light, and very occasionally decreased vision due to central visual axis sparing.

Prolonged and uncontrolled keratopathy may result in active scleritis, typically in the same or adjacent area. Clinical manifestations can vary in severity from mild to severe, from peripheral corneal thinning to stromal and ulcerative forms, requiring intensive treatment such as peripheral ulcerative keratitis (PUK) [79]. It is represented by ulcerations and tearing of the corneal epithelium, generally associated with limbus inflammation, iritis, and localized or diffuse scleritis. It is often unilateral but can affect both eyes. Recognizing and differentiating the specific ocular manifestations, is important to evaluate prognosis and therapy. For instance, research reported a prompt regression of corneal inflammation and depression of keratolysis in CD patients with PUK [80].

Additionally, infiltrates and scarring fibrosis process can damage all layers of the cornea, including the epithelium, stroma, and endothelium. [81]. For instance, Salzmann nodules, pearl-grey opacities, are a peculiar orbital complication in CD patients with a prior history of corneal inflammation. They result from the progressive partial or complete replacement of the Bowman’s layer by eosinophilic material. [82,83]. Severe ocular manifestations are rare and anecdotal. These include orbital inflammation such as dacryoadenitis, palpebral ptosis, lid margin ulcers, orbital myositis, and ocular myasthenia graves. Additionally, there are vascular disorders such as retinal vasculitis (including central serous chorioretinopathy, serpiginous chorioretinopathy, acute macular neuroretinopathy, and macular edema) or retinal vascular occlusions and neovascularization, as well as neurological affections such as optic neuritis and pseudotumor cerebri [46,49]. Demyelination and ischemic damage are the most common causes of optic nerve disorders [84]. Optic neuritis (ON), also known as retrobulbar neuritis, is a serious visual condition caused by the demyelination and degeneration of the optic nerve. This results in inflammatory edema and injury of the myelin sheath surrounding the nerve with potentially irreversible axonal damage and neurological impairment. [85]. Although rare, it is important to understand the pathogenesis ON and its association with IBD to prevent irreversible vision loss and impairment of quality of life. [86]. ON is often linked to other autoimmune disorders, particularly multiple sclerosis (MS), which has a higher prevalence in IBD patients compared to controls [87]. Furthermore, it is worth noting that ON may precede MS and that the risk of developing full-blown MS accumulates to 50% within 15 years [88]. Additionally, demyelinating disorders, such as ON, have been associated with TNF inhibitors, although a definitive cause–effect relationship has not been established [89,90,91,92]. However, ON during TNF inhibitor treatment is rare, and it has been reported to occur with comparable frequency among patients exposed to disease-modifying antirheumatic drugs (DMARDs) [93]. In a multicenter retrospective study conducted on IBD patients, it was found that ON during anti-TNF therapy was not statistically more prevalent compared to patients with ON off therapy. Additionally, a family history of ON or MS resulted as a risk factor for developing demyelinating disorders. Therefore, the use of TNF-inhibitors must be approached with caution among these patients, and ophthalmologic and neurological follow-up is recommended [94].

In Figure 2A the clinical presentation of episcleritis and anterior scleritis is displayed with vascular congestion and hyperemia; anterior uveitis (in Figure 2B) presents with non-granulomatous endothelial infiltrates, and irido-lenticular synechiae with subsequent pupillary alteration; Figure 2C shows an optic disc swelling (papilledema) and ischemia at retinal fluorescein angiography; feather- or flame-shaped hemorrhages and associated vascular congestion of the optic disc are shown in Figure 2D.

## 7. Treatment

Management of o-EIMs is complex and challenging; concomitant EIMs and IBD extent and severity are relevant factors to be considered when making treatment decisions and establishing prognosis. The current treatment options are summarized in Table 1. Episcleritis is generally a self-limited manifestation that resolves with control of IBD flares and a conservative approach [46,47,62]. Artificial tears or other topical lubricants are preferred rather than topical nonsteroidal anti-inflammatory drugs (NSAIDs) [95]. Oral NSAIDs are commonly used for episcleritis [96], but their use must be cautious because of some uncertainties regarding safety in IBD patients [97]. Topical glucocorticoids must be reserved for patients who do not respond to conventional treatment [62].

Conversely, scleritis and uveitis require immediate referral to an ophthalmologist. The initial treatment of scleritis is oral NSAIDs in a stepwise approach, followed by oral systemic corticosteroids (1–1.5 mg/kg/day), especially in case of a high risk of vision loss [51]. Posterior and/or necrotizing scleritis may require intravenous corticosteroid therapy [98]. Corticosteroid injections are recommended, usually in combination therapy [99] for both scleritis and uveitis. In particular, topical cycloplegics are suggested in combination with topical steroid treatment in anterior uveitis [46,54,62]. Conversely, intermediate, posterior and panuveitis require systemic steroid treatment from the beginning due to the low penetrance of topical agents in the posterior eye sector [100]. Intraocular implants with dexamethasone or fluocinolone could be an effective alternative [101,102], minimizing the dose or frequency of administration, but at the same time exacerbating ocular side effects such as cataract formation, ocular hypertension, and glaucoma or chorioretinopathy [103].

Although anti-metabolites (e.g., azathioprine), calcineurin inhibitors (e.g., cyclosporine), or alkylating agents (e.g., cyclofosfamide) have historically been used as a second-line therapy for ocular inflammation that does not respond to steroid treatment [49,104], the advent of biologic therapy has completely revolutionized the treatment of o-EIMs and has become the first choice for patients who do not respond.

Since the early 2000s, tumor necrosis factor inhibitors (TNF-inhibitors), such as infliximab (IFX) and adalimumab (ADA), have demonstrated high efficacy in the treatment of EIMs [105,106]. For instance, IFX was associated with a strong and long-term remission rate of ocular inflammation in an 8-year prospective cohort study [107,108]. ADA is now considered the first-choice TNF-inhibitor thanks to three multinational consecutive randomized, double-masked, placebo-controlled trials involving patients with uveitis (Visual I–III). In Visual I, Jaffe et al. reported almost twice as much time to treatment failure in patients undergoing ADA compared to the prolonged steroid therapy group [109], and its efficacy was higher in various forms of refractory uveitis [110]. Furthermore, ADA was associated with better prevention of uveitis flares and a decrease in vision impairment in Visual II, compared to steroids [111]. Lastly, the safety and efficacy of extended ADA treatment were assessed in Visual III; ADA was effective not only in achieving clinical uveitis’ remission and in guaranteeing a prolonged higher steroid-free quiescence, but also it was associated with an enduring prevention of ocular inflammation flares [112]. Further recent evidence comes from a Swiss cohort study of 224 IBD patients that reported a uveitis response rate of 72.5% in patients treated with TNF-inhibitors [113]. In addition, a recent systematic review and meta-analysis reported similar results, albeit with a lower level of evidence due to the smaller number of patients considered compared to other EIMs such as arthritis or cutaneous manifestations [114,115,116]. Indeed, TNF-alpha plays an important role in ocular inflammation by recruiting leukocytes and modifying the vascular endothelium and the blood–retinal barrier [117]. Further evidence on the importance of the TNF pathway in ocular inflammation in IBD is provided by the TNF-like cytokine (TL1A) and its receptors (DR3 and DcR3), which are upregulated in IBD patients and induce intestinal and ocular inflammation; TL1A blockade in mice was associated with a reversal of intestinal damage and a reduction in extraintestinal involvement [118].

Additional immune pathways may be associated with the onset of ocular inflammation and may become therapeutic targets. For example, IL23/17 plays an important role in the differentiation and stimulation of autoimmunity in both the gut and the eye [119]. Anti-IL23/17 (e.g., Ustekinumab) is now approved for IBD and has been successfully used for uveitis [120]; it may be a valid alternative in recurrent or refractory ocular inflammation, although more data are needed [100]. Since the current role of IL-17 in non-infectious ocular inflammation is well known, anti-IL-17 (e.g., Secukinumab) has been successfully used in various forms of uveitis [121]; however, anti-IL-17 has been associated with IBD exacerbation and intestinal inflammation through not fully understood pathogenetic mechanisms [122]; more data are needed to assess safety in IBD patients [123]. The use of biologics has revolutionized the management of IBD. They have demonstrated a higher safety and efficacy profile, providing long-term control of inflammatory ocular disease, as previously reported. Side effects, particularly infectious diseases, are rare and mostly mild. There is no higher incidence of skin tumors or the potential risk of developing eye melanoma in the population affected by o-EIMs compared to patients with no extra-intestinal inflammation extension [89]. The major limitation of this type of drug is still poor compliance resulting from parenteral administration.

Finally, promising therapies include not only biologics but also small molecules, which have the advantage of being orally administered. One example is the Janus kinase (JAK) inhibitors. JAKs are important proteins that mediate the intracellular signaling of several cytokine receptors, including INF-y and interleukin 2 (IL-2) [100,124]. Tofacitinib has been successfully used to treat two patients with refractory scleritis and uveitis [125]. For its inhibitory effect on several JAK proteins involved in multiple cytokine signaling, tofacitinib could potentially act at different moments in the pathogenesis of EIMs; however, since specific immune pathways related to the management of o-EIMs are still undetermined, further data are required to verify its potential role in the management of ocular inflammation.
cells-13-00142-t001_Table 1Table 1This is a table summarizing epidemiology, pathogenesis, clinical aspects, and treatment of ocular manifestations in IBD patients.
PathophysiologyClinicalManifestationsEpidemiologyTherapeuticMoleculesEpiscleritis/scleritisMultifactorial, mostly unknown¬Acute hyperemia¬Irritation¬Burning and tenderness¬2–5% of IBD patients¬Strongly related to intestinal flares¬Treatment of IBD flare¬Topical steroidsUveitis Ectopic over-expression of adhesion molecules and chemokines in extra-intestinal tissues *Bacterial antigen cross-reactivity molecular mimicryMicrobial translocation¬Conjunctival injection¬Photosensitivity ¬Blurry vision or headache¬Light sensitivity, scotomas, potential acute blindness (in case of posterior uveitis)¬0.5–3.5% of IBD patients¬CD >> UC¬Bilateral, insidious onset, and persistent if associated with CD ¬Topical/systemic steroids¬Cyclosporine (2nd line)¬Anti-metabolites (2nd line)¬Anti-TNFs (2nd line)¬Vedolizumab ineffective¬Uncertain efficacy of Tofacitinib and Ustekinumab (case reports)KeratitisAltered innate Immunity¬Corneal ulceration¬Eye pain¬Foreign body sensation¬Rare (1.13%) [126]¬Usually secondary to scleritis¬Systemic and topical steroidsRetinal vasculitis, retinal vascular occlusions, optic neuritisGenetic predisposition **Environmental factors
¬Rare (case reports)¬Depending on the cause, surgery can be needed* Demonstrated in mice models of uveitis. ** [HLA]-B*27, B*58 and HLA-DRB1*0103 have been associated with uveitis in IBD populations. IBD: inflammatory bowel diseases; CD: Crohn’s disease; UC: ulcerative colitis.

## 8. Conclusions

At present, O-EIMs represent a major challenge to clinicians, not only in terms of diagnostic pathways, but also in terms of therapeutic choices. O-EIMs are less common in IBD patients compared to skin and musculoskeletal manifestations; however, non-ophthalmological specialists should be able to recognize them due to their high morbidity and potential risk of vision loss. The onset of O-EIMs is insidious and may occur during IBD flares or completely independently of disease activity, and occasionally precedes intestinal inflammation. The pathogenesis is still poorly understood and is mainly attributed to expansion of intestinal inflammation due to ectopic chemokine expression with subsequent T cell migration. However, genetic predisposition, pre-existing immune dysregulation and dysbiosis may independently determine a pro-inflammatory state that alone can explain the onset of o-EIMs. In conclusion, O-EIMs’ pathogenesis and management are not fully understood; current limitations to better clarify underlying mechanisms are represented by hurdles in reconstructing adherent animal models, the lack of clinical trials specifically designed to evaluate ocular manifestations in IBD and, most of the time, EIMs are considered exclusion criteria in IBD drugs clinical trials. However, single-cell RNA techniques may assume a promising role in defining EIM inflammatory processes leading to future new treatment perspectives [94].

Although corticosteroids still play an important role in the treatment of o-EIMs, their chronic use is limited by their systemic side effects and ocular complications. Therefore, immunomodulators and biologics, especially TNF inhibitors, are currently considered the backbone of the management of o-EIMs, ensuring effective and timely interventions, especially in refractory and recurrent manifestations. Indeed, ADA was the first anti-TNF approved for intermediate, posterior, and panuveitis. As already extensively discussed in rheumatology [127], not all available treatments for IBD are equally effective for o-EIMs, particularly for uveitis, and for some classes of treatment a discrepant efficacy across o-EIMs has been observed. RCTs specifically addressing o-EIMs and conducted in IBD patients with these conditions are currently lacking. Despite therapeutic algorithms, most patients with o-EIMs need a personalized approach Ongoing research and a deeper understanding of the mechanisms underlying ocular manifestations would allow new targeted therapies and personalized treatments for IBD patients. Given the continuing clinical challenges and the higher risk of IBD patients with one EIM to develop others in another district, a multidisciplinary approach (between gastroenterologists, rheumatologists, dermatologists, and ophthalmologists) is paramount. Awareness of extraintestinal manifestations among IBD clinicians is essential for timely recognition and prompt referral to specialists.

## Figures and Tables

**Figure 2 cells-13-00142-f002:**
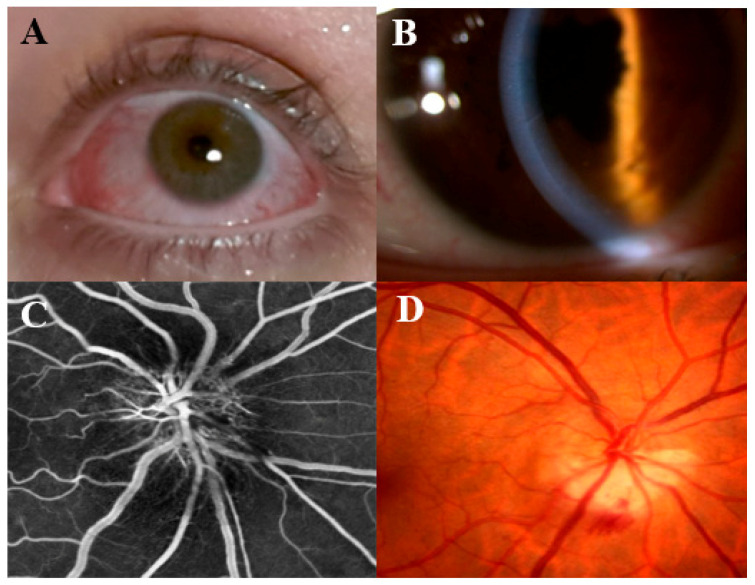
Clinical manifestations of episcleritis/scleritis, anterior uveitis, and retinal ischemia. In panel (**A**), the clinical presentation of episcleritis and anterior scleritis is displayed with vascular congestion and hyperemia; anterior uveitis (in panel (**B**)) presents with non-granulomatous endothelial infiltrates, and irido-lenticular synechiae with subsequent pupillary alteration; panel (**C**) shows an optic disc swelling (papilledema) and ischemia at retinal fluorescein angiography; feather- or flame-shaped hemorrhages and associated vascular congestion of the optic disc are shown in panel (**D**).

## Data Availability

Data sharing is not applicable. No new data were created or analyzed in this study.

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
