# Peer review of "Ophthalmological Manifestations in Inflammatory Bowel Diseases: Keep an Eye on It"

_cells, 2024, doi:10.3390/cells13020142_

Round 1
Reviewer 1 Report
Comments and Suggestions for Authors
The topic is relevant in the field of IBD as it concerns up to 13% of IBD patients lowering their quality of life. This review is very informative describing extensively also the pathogenesis of ocular extraintestinal manifestations in IBD. The references are ppropriate. The figures are excellent. This review aims to summarize and clarify ocular extra-intestinal manifestations (O-EIMs') characteristics, and clinical manifestation with a particular focus on the molecular pathogenesis, as well to discuss current and future management perspectives. This is a very well-written and comprehensive review. I have only two minor points.
Minor points
1. Please clarify the type of this review (narrative, comprehensive, systematic, etc.) and the key words that were used for literature searching.
2. The legent of Table 1 should be revised. Regarding the table, prevalence of each o-EIM could be added.
Author Response
The topic is relevant in the field of IBD as it concerns up to 13% of IBD patients lowering their quality of life. This review is very informative describing extensively also the pathogenesis of ocular extraintestinal manifestations in IBD. The references are appropriate. The figures are excellent. This review aims to summarize and clarify ocular extra-intestinal manifestations (O-EIMs') characteristics, and clinical manifestation with a particular focus on the molecular pathogenesis, as well to discuss current and future management perspectives. This is a very well-written and comprehensive review. I have only two minor points.
Re: Thank You for Your comments, we are glad that You appreciated our paper.
- Please clarify the type of this review (narrative, comprehensive, systematic, etc.) and the key words that were used for literature searching.
Re: Thank you for Your comment. We specified that the paper is a narrative review and we added the information requested; in particular, we added a specific paragraph (2. Materials and methods) regarding your last suggestion
- The legend of Table 1 should be revised. Regarding the table, prevalence of each o-EIM could be added.
Re: Thank you for Your comment. We revised the description and legend of Table 1, as suggested. We added the prevalence of keratitis associated with IBD; retinal vascular occlusions, vasculitis or optic neuritis are rarer and anecdotal events with no clear prevalence.
Reviewer 2 Report
Comments and Suggestions for Authors
The paper is very interesting and actual but the Authors should add a section about the ocular side effects of biological therapies involving particularly the Optic Nerve
1) The main question is a larger diffusion of knowledge of ocular manifestations in IBD
2) the topic is relevant and actual
3) a more accurate development of ocular manifestations in IBD
4) I think that the methodoly is correct
5) the conclusions are consistent with data
6) the references are appropriate
Author Response
The paper is very interesting and actual, but the Authors should add a section about the ocular side effects of biological therapies involving particularly the Optic Nerve
- The main question is a larger diffusion of knowledge of ocular manifestations in IBD
- the topic is relevant and actual
- a more accurate development of ocular manifestations in IBD
- I think that the methodogy is correct
- the conclusions are consistent with data
- the references are appropriate
Re: Thank you for Your comments. We are glad that You found our paper clinically relevant and well-discussed. We included optic neuritis related to biological therapies in our papers, as You suggested.
Reviewer 3 Report
Comments and Suggestions for Authors
The manuscript by Giulia Migliorisi et al. summarized the ophthalmological manifestations in inflammatory bowel diseases. The study is of interest to the readers and I have the following comments and suggestions:
1, current limitations and future directions in the study ocular EIMs must be discussed.
2, why ocular EIMs (O-EIMs) has a higher prevalence in Crohn's Disease but not UC? Potential mechanisms should be discussed.
3, what are the current drugs for the treatment of ocular EIMs? Advantages and limitations should be discussed.
Author Response
The manuscript by Giulia Migliorisi et al. summarized the ophthalmological manifestations in inflammatory bowel diseases. The study is of interest to the readers, and I have the following comments and suggestions:
Re: Thank you for Your comment. We appreciate Your suggestions.
- Current limitations and future directions in the study ocular EIMs must be discussed.
Re: Thanks for Your comment. Current limitations and future perspectives of ocular EIMs’ evaluation and study have been added in the conclusion paragraph.
- Why ocular EIMs (O-EIMs) has a higher prevalence in Crohn's Disease but not UC? Potential mechanisms should be discussed.
Re: Thanks for Your suggestion. We added a brief discussion in the pathogenesis paragraph, as kindly suggested.
- What are the current drugs for the treatment of ocular EIMs? Advantages and limitations should be discussed
Re: Thanks for Your comment. We implemented the treatment paragraph, reporting current therapies' major advantages and limitations, especially for biologics.